# A Review of the Current Status of Gestational Diabetes Mellitus in Australia—The Clinical Impact of Changing Population Demographics and Diagnostic Criteria on Prevalence

**DOI:** 10.3390/ijerph17249387

**Published:** 2020-12-15

**Authors:** Josephine G Laurie, H. David McIntyre

**Affiliations:** 1Department of Obstetric Medicine, Mater Mothers’ Hospital Brisbane, Queensland and Mater Clinical Unit, The University of Queensland, South Brisbane, QLD 4101, Australia; 2Department of Obstetric Medicine, Mater Mothers’ Hospital Brisbane, Queensland and Mater Research Institute, The University of Queensland, South Brisbane, QLD 4101, Australia; david.mcintyre@mater.org.au

**Keywords:** gestational diabetes mellitus, epidemiological impactors, prevalence, remote, diverse, model of care, COVID-19, obesity, maternal age, ethnicity

## Abstract

The current status of gestational diabetes mellitus in Australia reveals an almost quadrupling prevalence over the last decade. A narrative review of the current Australian literature reveals unique challenges faced by Australian maternity clinicians when addressing this substantial disease burden in our diverse population. Rising rates of maternal overweight and obesity, increasing maternal age and the diversity of ethnicity are key epidemiological impactors, overlaid by the 2015 changes in screening and diagnostic parameters. Our vast land mass and the remote location of many at risk women requires innovative and novel ideas for pathways to diagnose and effectively manage women with gestational diabetes mellitus. By modifying and modernizing models of care for women with gestational diabetes mellitus, we have the ability to address accessibility, resource management and our acute response to global events such as the COVID 19 pandemic. With continuing research, education and robust discourse, Australia is well placed to meet current and future challenges in the management of gestational diabetes mellitus.

## 1. Introduction and Background

Gestational diabetes mellitus (GDM) is one of the most common medical complications of pregnancy in Australia. The increase in maternal overweight and obesity, advancing maternal age, “at risk” ethnic group representation and the change in diagnostic processes and criteria (summarized in Table 1) from the previous Australasian Diabetes in Pregnancy Society (ADIPS) criteria [1] to those recommended by the International Association of the Diabetes in Pregnancy Study Groups (IADPSG) and endorsed in 2013 by the World Health Organization [1] have all contributed to a dramatic increase in the prevalence of GDM in Australia.

The aim of introducing more widespread testing and uniform criteria for GDM was to improve maternal and foetal outcomes, as suggested by the landmark Crowther [3] and Landon [4] randomized controlled trials (RCTs). However, increased rates of GDM diagnosis may also contribute to potential physical and psychological morbidity for the mother and physical morbidity for the child and unfortunately, this may be compounded by an unmet resource gap, as resources may not be available to provide “best practice” care for all GDM women.

There are both immediate and long-term clinical and health economic advantages and burdens which need to be clearly balanced when considering GDM [5]. The balancing of needs and outcomes provides an opportunity for innovative thinking around how we deliver care and how we acutely respond to unexpected world events, such as the current COVID-19 pandemic [6].

The authors conducted PubMed and Medline searches using the terms: GDM, Australia, screening, models of care and postnatal follow up to gather information for this review. Articles were selected to specifically represent Australian cohorts and data for an Australian perspective. Australian prevalence data were drawn directly from the National Diabetes Service Scheme (NDSS) (which registers all Australian residents with a diagnosis of gestational diabetes who receive access to government subsidies for the purpose of diabetes care) and the Australian Institute of Health and Welfare website. [7]

This review examined the prevalence of GDM in Australia since the adoption of the IADPSG diagnostic criteria (officially 1 January 2015, but variable across the country). Universal screening is recommended for pregnant women, not previously diagnosed with hyperglycaemia, and between 24 and 28 weeks gestation [8]. This highlights the distinctive local Australian challenges of delivering care to geographically isolated women across vast distances and the specific needs and concerns of our First Nation women. This reviews the current models of care and considers how we might cater for women with cultural and linguistically diverse backgrounds. Finally, it addresses the current COVID-19 screening situation and the need for improved post-natal care.

## 2. Prevalence

A recent position statement released by Diabetes Australia [9] reports that almost 41,000 Australian women were diagnosed with GDM in 2019 (refer to Figure 1 below), one fifth of whom presenting with a repeat diagnosis. Approximately 300,000 babies were born in Australia in 2018 [10], so this equates to around 14% of Australian gravidas being currently classified as having GDM. There is, however, enormous variability between regions in Australia which is discussed further below. International prevalence data drawn from Brown et al. [11] in 2017 also show the high variability of GDM rates ranging from single digit prevalence in Japan, to over 25% of pregnancies affected in California and greater than 45% in the United Arab Emirates.

With a doubling of the number of affected women in the last decade, GDM is the fastest growing subtype of diabetes in Australia [9]. The rapid increase in the prevalence of GDM is multifactorial and complex, with high variability based on ethnic representation, obesity rates and maternal age. Almost half of pregnant women in Australian are now classified as being overweight or obese. Cheney et al. [12] addressed the contribution of rising maternal BMI (Body Mass Index kg/m^2^) to GDM by analysing a large data subset of 9245 women from a single maternity centre in Sydney during the period 2010–2014. Cheney found that the population attributable fraction (PAF) for GDM attributable to women being overweight or obese was 17%.

In the last two decades, the average maternal age has also increased by more than 4 years and the number of women with ethnic backgrounds at a higher risk of GDM (in particular those born in southern and central Asia, Southeast Asia, North Africa and the Middle East) has doubled [7].

## 3. Impact of Changes in Screening and Diagnostic Criteria

A change in diagnostic criteria has also had a significant impact and continues to be a point of contention and consideration in Australia. Traditionally in Australia, GDM was diagnosed using a fasting 75 g oral glucose tolerance test (OGTT), with or without prior non-fasting “glucose load” testing. As summarized in Table 1, diagnostic thresholds of ≥5.5 mmol/L fasting and ≥8.0 mmol/L at 2 h on the OGTT were set empirically by an “ad hoc working party” in 1991 [2] and although their evidence base was thin, these criteria were widely used from 1991 to 2015.

Following the Hyperglycemia and Adverse Pregnancy Outcome (HAPO) study findings, the IADPSG proposed new diagnostic criteria in 2010 [13]. These were endorsed by the World Health Organisation (WHO) [1] and the Australasian Diabetes in Pregnancy Society (ADIPS) in 2014 [14]. These “IADPSG/WHO2013” criteria were adopted by more than 90% of Australian maternity providers and have been in use in the authors’ home state of Queensland since January 2015 [15].

In previous as well as current diagnostic criteria for GDM in Australia, all values relate to glucose concentrations at the oral glucose tolerance testing (OGTT) before or after the ingestion of 75 g of glucose in the fasting state. The ADIPS 1991 guidelines recommended two-step testing with an initial non-fasting glucose load (50 or 75 g). A full OGTT was recommended if venous plasma glucose was ≥7.8 mmol/L 1 h after a 50 g load or ≥8.0 mmol/L 1 h after a 75 g load. The current criteria do not recommend two-step testing.

Whilst a change in GDM diagnostic criteria had a significant impact in specific patient cohorts, Figure 1 demonstrates that a national upward trend in GDM prevalence was already evident prior to the implementation of the IADPSG guidelines. This trajectory has continued after the change to IADPSG/WHO2013 diagnostic criteria post-2015.

A recent publication by Brown et al. [16] reviewed 112,308 confinements over 10 years in an ethnically diverse health district in Sydney. Interestingly, their data showed a sharp increase in the prevalence of GDM in 2013, before changes to diagnostic processes and criteria were implemented. This was particularly evident in women of South Asian origin and was unrelated to changes in BMI in that cohort of women over the study period.

In a publication in 2016, Moses et al. [17] assessed the impact of the IADPSG/WHO2013 criteria on GDM prevalence using public and private maternity patient data in southern New South Wales. They concluded that the new criteria resulted in one-third more women diagnosed with GDM when compared with the previous ADIPS criteria. Elevated fasting glucose contributed to the majority (56%) of GDM diagnoses.

The local impact of the screening changes has been measured and published from several other metropolitan and regional centres and their research findings are summarised below.

In 2018, a published retrospective audit conducted at Townsville Hospital [18], a regional centre in Queensland, considered the impact of changing to the IADPSG diagnostic criteria in 2015. This analysis was performed using patient data for 6 months in 2014 pre-adoption and again in the 2015 post-adoption phase. The authors reported a doubling in the incidence of GDM using the new IADPSG diagnostic criteria (from 9.8% pre- to 19.6% post-adoption, *p* < 0.01).

Similarly, Cade et al. [19] performed a pre- and post-change of criteria assessment in Melbourne, Victoria. The incidence of GDM rose from 5.93% pre-IADPSG adoption to 10.3% post-adoption. They did not show an improvement in the maternal or neonatal outcomes post-adoption and demonstrated an increased net cost of care due to the substantially increased number of women diagnosed with GDM.

In addition, Cheung et al. [20] also published 2018 a retrospective cohort study which was performed in Western Sydney pre- (2014) and post- (2016) the implementation of universal screening, based on the ADIPS 1991 [2] criteria which were restated in 1998. A comparison was then made between the ADIPS1998 criteria and the IADPSG criteria reviewing prevalence, perinatal outcomes and the independent influence of obesity in this cohort. The reported outcomes included: an increase in prevalence of only 2.7% (very modest when compared with co-located health districts), a significant and independent association between maternal obesity and adverse pregnancy outcomes (regardless of which diagnostic criteria were used) and the relative risk of ethnicity on the positivity of screening. They found a higher risk of GDM using IADPSG criteria in women of Asian and Middle Eastern background, with lower risk noted for women of North and South America and European background.

Similarly, Wong et al. [21] looked at GDM prevalence variation due to ethnicity and obesity in their culturally diverse maternity population in Western Sydney. Their findings showed an overall doubling of prevalence in GDM using the IADPSG criteria (14.5 to 29.6%), however, in the subgroup analysis, the prevalence in women from East/South-East Asia increased the least (19.2 to 22.3%) and the prevalence in women from South Asia the most (22.0 to 44.4%). The prevalence of GDM in obese women (BMI > 30 kg/m^2^) was substantial at 45.9%.

A recently published Queensland retrospective cohort study [22] of GDM prevalence and pregnancy outcomes pre- and post- the implementation of the (IADPSG) diagnostic criteria revealed that GDM rates increased state-wide from 8.7 to 11.9% post-implementation. This report assessed the perinatal data the year before (2014) and the year following (2016) Queensland’s move to one step screening and diagnosis of GDM according to IADPSG criteria. Each historical cohort included more than 60,000 women. Perinatal outcomes of interest were defined as gestational hypertension, caesarean birth, gestational age at delivery, birthweight, preterm delivery, large-for-gestational age (LGA) neonates, small-for -gestational age (SGA) neonates, neonatal hypoglycaemia, and respiratory distress. The authors reported a very small decrease in neonatal respiratory distress without significant changes in other perinatal outcomes post-implementation. The authors considered positive impactors including an (independent) improved detection of neonatal hypoglycaemia during this time period, and negative impactors such as increased maternal BMI and the inadequate resourcing of care as possible confounders for the lack of discernible improvement in neonatal outcomes. However, we note that assessing overall outcomes on a population basis is unlikely to show significant changes. In this report, the absolute increase in GDM frequency was 3.2% and only these women would have received different treatment in the two study periods. By contrast, around 88% of women would have been considered as “non GDM” in both study periods and would thus have received similar care either before or after the changes in diagnostic processes and criteria were implemented.

Predictably, the direct impact of the introduction of the IADPSG criteria on GDM prevalence was influenced by several factors. Rapid changes in the ethnic diversity of childbearing women in certain regions, increasing maternal age and maternal BMI will have a variable impact depending on which population is assessed. The impact will fluctuate between regional, remote and metropolitan areas across the nation. This substantial variability adds to the complexity of screening but also imposes an inequitable burden and challenge in delivering GDM care for some health districts.

## 4. Australia Specific Groups and Issues

Australia has some unique challenges in managing GDM ranging from the geographical location of “at risk” women, to screening procedures, and remote treatment strategies. Regarding “at risk” women, First Nation mothers are not only more likely to develop GDM than non-indigenous women, but also have far higher pregnancy complication rates.

The Pregnancy and Neonatal Diabetes Outcomes in Remote Australia (PANDORA) study [23], an observational study in the Northern Territory, has revealed high rates of diabetes with a preponderance of pre-gestational diabetes in indigenous women, when compared with their non-indigenous counterparts. In this cohort, indigenous mothers were younger and had higher smoking rates, and in the GDM/DIP (diabetes in pregnancy) subgroup, they suffered poorer birth outcomes with an increased proportion of large for gestational age (LGA) babies (19 vs. 11%).

Due to Australia’s immense land mass, women who reside in remote and rural areas have limited access to diagnosis, education and specialist management of GDM. Their “at risk” status due to geographic isolation is compounded by indigenous or another “at risk” ethnicity, higher rates of elevated BMI and social disadvantage including limited access to fresh food and healthy eating options [24].

Following the identification of and access to these “at risk” remote population, the next barrier is accurate laboratory testing. Between 2015 and 2018, the Optimisation of Rural Clinical and Haematological Indicators for Diabetes in Pregnancy (ORCHID) study [25] cohort was recruited in rural Western Australia. This study was conducted to address concerns about geographically mandated long delays between the time of venesection at various remote sites to the centralized laboratory facilities, focusing on the issue of pre-analytic glucose degradation (due to in-sample glycolysis). There were concerns that this could lead to the under diagnosis of GDM in an “at risk” group. The authors assessed the potential impact of alternate pre-analytical protocols to minimise glycolysis. Most OGTT samples were stored at room temperature and 92% (461/501) had a substantial delay before analysis; median 5 h, range 2.3–124 h. The authors estimated that 62% of potential GDM diagnoses were missed amongst regional, rural and remote Western Australian women by the 24–28 week OGTT. This is due to the suboptimal sample processing issues, with the greatest pre- analytical error relating to the measurement of fasting venous plasma glucose. In a small subset of their cohort, they demonstrated that changing their pre-analytic sampling procedures, by storing samples in an ice–water slurry (following the protocol used in the HAPO study), was able to prevent much of the in-sample glycolysis. However, this was considered impractical for routine clinical use. By contrast, the use of fluoride citrate tubes without cooling was also able to prevent in sample glycolysis, but led to a systematic increase in laboratory glucose measurements of 0.2 mmol/L. This in turn led to an approximately doubled GDM prevalence, predominantly due to elevations in fasting glucose values. They suggested that laboratories could potentially correct for this difference by the systematic subtraction of 0.2 mmol/L from their analytic results, giving results comparable with those obtained with HAPO-aligned methods.

Concern regarding the pre-analytical handling of blood for OGTTs was also raised by a group in the Australian Capital Territory, with no geographical barriers. These authors reported a prevalence difference of 9% (20.6 vs. 11.6%) between protocols using early centrifugation vs. delayed centrifuge of glucose samples from the OGTT [26].

Thus, there is both a major challenge and a potential opportunity to define and implement better pre-analytical handling of OGTT specimens both remotely and regionally in Australia to improve diagnostic accuracy.

Once identified and diagnosed, women with GDM then have variable access to expert care dependent on their geographical location. To address this concern, the Australian health system needs to focus on innovative technological solutions including opportunities to provide more care via digital platforms and Telehealth. In recent practice, such platforms were rapidly deployed and effectively utilised in many regions during the COVID-19 pandemic lockdown period for both family and specialist medical care. Research into the use and efficacy of digital solutions in Australia is lacking and an opportunity exists to further develop the potential use of these techniques in routine clinical practice.

## 5. Models of Care

Increased rates of GDM carry an obvious physical and emotional burden for the affected women themselves, as well as higher risks of immediate and longer-term morbidity for both the mothers and their babies and the health economic burden for the management of the affected dyad. The other significant burden falls to the health practitioners trying to provide optimal care for this ever-increasing cohort of women. Traditional models of care for women with GDM are not sustainable, with the major limitations being inadequate funding and inadequate numbers of trained health care providers. Unfortunately, additional funding and staffing has not been forthcoming in the current economic climate and due to the variability of population demographic, management solutions are unlikely to be generalisable.

Sina et al. [27] undertook a thematic analysis from 15 volunteer diabetes in pregnancy services from Australia and New Zealand in 2018. Their report highlights not only the difficulty of dealing with increased patient numbers in a traditional model of care, but also the complexity of delivering appropriate care in remote, rural, urban and metropolitan settings and for women with culturally and linguistically diverse (CALD) backgrounds. The prevalence of GDM in this analysis ranged between 10.4% in metropolitan Victoria to 26.9% in peri-urban Queensland.

All clinics reported that they provided initial group education classes for GDM women run by a credentialed diabetes educator and dietitian and a subsequent individual session for each patient. Commonly, women were provided with a contact phone number to call if the blood glucose levels were out of range. At this point, and no doubt due to the workload, most clinics provided a “step-up” option, where only women with self-monitored blood glucose (SMBG) out of range receive ongoing care from a diabetes educator or endocrinologist. This resulted in most women remaining with their primary antenatal team, which has some potential benefits in terms of the continuity of care. However, either the primary maternity provider or the woman herself must then take the role of a diabetes advocate—seeking an escalation of care when required. Experience tells us that often the women with the least health literacy or self-advocacy have the greatest need for care. In light of this, models of care need to accommodate all women including those with literacy and language difficulties, those with adherence concerns and women with accessibility disadvantages. From the thematic analysis reported in this paper, common challenges included: busy clinics, lack of coordination between obstetric and diabetes care, CALD, large numbers of GDM women requiring insulin and difficulties in delivering services to patients from rural and remote areas. Proposed solutions included: alteration in traditional models of care, step-up and step-down models, interdisciplinary and midwifery champion approaches, on site interpreters, and technology solutions including portals and telehealth.

Digital monitoring and telehealth provision of care have distinct advantages in overcoming some of the geographical and workflow barriers for delivering care. Studies dating back 10 years report high patient acceptability for web-based interventions in GDM. Despite their low-income status, most women had and have access to smart phones and/or computers and report this as their preferred medium for learning. [28] Mobile (smartphone) health solutions are a suitable option for GDM care due to the nature of monitoring, transmitting blood glucose readings from a meter, to phone, to a clinician portal. Education can also be provided digitally in the form of video or teleconferencing mediums. Such a web-based educational intervention was trialled in a multi-ethnic cohort in metropolitan Melbourne [28] with an intervention aimed at a low level of literacy but delivered in English. The three domains assessed were knowledge of GDM, food values and GDM self-management principles. The authors reported the intervention as successful in the first domain with over 70% of women improving their knowledge scores post-intervention. The sample size was small (*n* = 21) and the authors commented on the comprehension issues with the questionnaire relating to the second and third domains. Language-specific education was proposed for a subsequent trial.

## 6. Treatment Challenges

The next challenge is how to meet the treatment needs of GDM women. The time-critical nature of pregnancy care means that therapy must be delivered promptly and the highly variable population demographics of pregnant women in Australia mean that the treatment must be adapted to a wide range of cultural and linguistic contexts. Medical nutrition therapy (MNT) remains the foundation of the initial and in many cases ongoing management of GDM. The aims of MNT include meeting the nutritional needs of the mother and baby, and the attainment and maintenance of glycaemic targets.

However, an important risk factor for GDM is ethnic origin. More than 25% of the Australian population are first generation migrants, 7.6% of Chinese origin. Of birthing mothers, more than a third were born overseas, most commonly from India, China and New Zealand [10].

As the foundation of GDM care, MNT education must therefore be culturally appropriate for multiple ethnic groups. As previously mentioned, all the models of care surveyed in Australasia used group education for teaching on GDM and MNT. Wan et al. [29] performed a mixed methods study in metropolitan Melbourne comparing the perceptions and experiences of dietary self-management and nutritional needs of ethnic Chinese women (*n* = 44) and Australian born white women (*n* = 39). The authors discovered that while all the white participants expressed satisfaction with the GDM teaching, most first-generation migrant Chinese participants indicated a strong preference for detailed, prescriptive, brand-orientated advice. Of concern, half the Chinese participants engaged in restrictive eating behaviours to reduce errors and improve self-management. Twelve of the 44 Chinese women experienced hunger, compared with two of the 39 white women. 

A 2015 review article from Liverpool, Sydney [30], also highlighted the challenges of delivering MNT in a GDM service to women of CALD backgrounds. These authors proposed culturally sensitive treatment including tailored MNT and an individualised approach to insulin prescription, citing the significant variation in insulin requirement between ethnicities.

Considering differing potential approaches to GDM education, a cohort of GDM women was assessed in an ethnically diverse part of Sydney, pre-IADPSG adoption [31]. Of the 743 women included, approximately half received group education for MNT and half individual education appointments. The group education participants had a small but statistically significantly lower HbA1c at diagnosis but the groups were otherwise well matched for baseline characteristics. The content of the dietary education provided was the same for both group and individual education. Subsequently, these authors reported an increased requirement for insulin in the education group (42 vs. 34.6%) and multivariable logistic regression defined “group women” as a predictor of insulin use. Perinatal outcomes were similar.

Updated NDSS data demonstrate an overall proportion of insulin use of 33.8% in 2019. With approximately 14,000 women requiring insulin therapy per year, MNT interventions which reduce the requirement for pharmacotherapy are potentially very attractive and merit specific funding. The most commonly used oral hypoglycaemic medication for gestational diabetes mellitus in Australia is Metformin, however, its use varies widely between centres, with some preferring insulin therapy alone. The NDSS data do not provide metformin use as a separate measure (many women will be co-prescribed Metformin and insulin therapy) and therefore an accurate assessment of the Australia-wide prevalence of use is not currently available. Some individual centres have published their clinical experience using Metformin. The main focus of these articles pertains to neonatal and maternal outcomes following Metformin therapy. Maple-Brown et al. [32] in the Northern Territory have reported prevalence data of 42–48% Metformin use in indigenous women with GDM and Diabetes in Pregnancy (DIP) and 14–35% in non-indigenous women with GDM/DIP, in an observational study during the period 2012–2016.

## 7. Impacts of COVID-19

Compounding the challenges of increasing prevalence, maternal obesity, advancing maternal age and unmet and unmanageable clinical needs, another dramatic and unexpected influence has been the current worldwide COVID-19 pandemic. In line with multiple international bodies [6] and with the aim of reducing COVID-19 transmission by person to person contact, Australia temporarily revised the recommendations for GDM testing processes and criteria. The COVID-19 revisions included a two-step process with an initial fasting glucose followed by the selective use of OGTTs: fasting glucose of ≥4.6 mmol/L does not require further testing, ≥5.1 mmol/L diagnoses GDM. Moreover, 4.7 mmol/L–5.0 mmol/L requires a full OGTT [33]. Van Gemert et al analysed the proportion of potentially missed diagnoses using this system and 6 years of OGTT data from the Illawarra region, in southern New South Wales [34]. Overall, 16,263 OGTTs were analysed and 1992 patients were diagnosed with GDM, giving a prevalence of 12.2%. Of these GDM women, 29% had a fasting level ≤4.6 mmol/L and would have been missed by the revised COVID-19 screening practice. The authors extrapolated that a fasting glucose cut off of <4.0 mmol/L would be required for a 95% confidence in exclusion of GDM. However, beyond the question of simple GDM frequency, a further secondary analysis of HAPO data from a multi-ethnic cohort involving five HAPO centres has reported that complications including LGA infants and operative birth are less frequent in those women who achieve a GDM diagnosis due to elevated 1 or 2h OGTT glucose but have a fasting glucose <4.7 mmol/L [35]. Therefore, the use of fasting glucose as a first step in GDM assessment may be warranted even after the COVID-19 pandemic. We acknowledge that all changes in GDM diagnostic strategies in the context of the COVID-19 pandemic were made on an empirical basis and that they have not been recommended for ongoing utilization outside this context. Further assessment of the perinatal outcomes using any revised screening programme is required before changes are made to the recommended testing protocols.

## 8. Post-Natal Follow Up for GDM

A post-natal follow up OGTT for GDM women is recommended in Australia at 6–12 weeks post partum. The responsibility for this usually falls to the woman’s general practitioner, however, adherence to this testing protocol is known to be suboptimal. Due to their higher risk nature, GDM pregnancies are usually managed in a hospital-led maternity model. Therefore, the general practitioner may not have seen the woman in her pregnancy or even be aware of her diagnosis [36]. Poor follow-up rates are seen despite Diabetes Australia sending reminder letters upon registration of GDM, at 12 weeks post partum and annually thereafter. Many contributory factors and barriers have been proposed, including poor communication from the maternity team at discharge, a lack of patient awareness about the potential risk of type 2 diabetes and reassurance of GDM “resolving” post partum [37]. Communication issues are evident not only in metropolitan centres [37] but also in our remote communities, with similar issues highlighted in a qualitative review of diabetes in pregnancy care in the remote Northern Territory [38].

## 9. Conclusions

GDM prevalence is clearly increasing across Australia. Important factors contributing to this increase are universal testing, the increased representation of higher risk ethnic groups amongst pregnant women, older age at childbirth, rising rates of maternal overweight and obesity, the population increase in all types of diabetes and the change to “one-step” IADPSG/WHO2013 diagnostic processes and criteria. Australia continues to attempt to assess and address the expanding need for GDM management with particular reference to “at risk” women, those with limited access to expert care and those with CALD needs. Practitioners involved in the care of women with GDM continue to debate, test and collaborate in many ways in an attempt develop pragmatic solutions to improve GDM diagnosis, management and prevention. With innovative solutions, education and awareness, we are well placed to respond to the challenges of 2020 and beyond.

## Figures and Tables

**Figure 1 ijerph-17-09387-f001:**
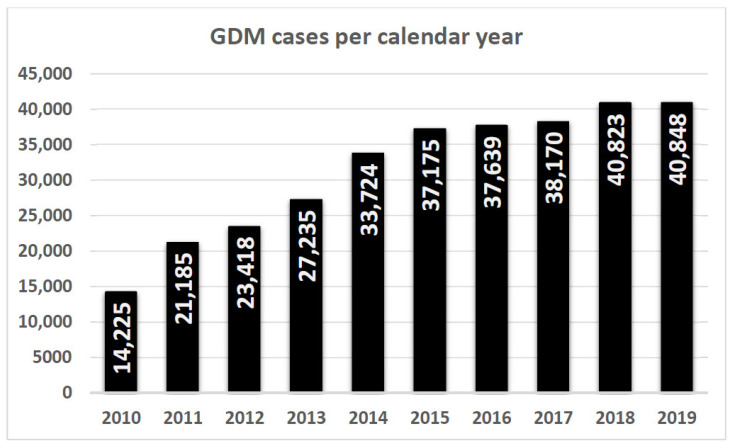
The number of women diagnosed with GDM in Australia per year (National Diabetes Service Scheme—NDSS) pre- and post-2015 IADPSG guideline implementation. Solid vertical columns refer to total GDM diagnoses enrolled with NDSS per annum.

**Table 1 ijerph-17-09387-t001:** Diagnostic criteria for gestational diabetes mellitus (GDM) in Australia. (ADIPS—Australasian Diabetes in Pregnancy Society; IADPSG—International Association of Diabetes in Pregnancy Study Groups).

Criteria	Fasting Glucose (mmol/L)	1 h Glucose (mmol/L)	2 h Glucose (mmol/L)
ADIPS 1991 [2]	≥5.5	N/A	≥8.0
IADPSG 2010/WHO 2013 [1]	≥5.1	≥10.0	≥8.5

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
