# Peer review of "A Review of the Current Status of Gestational Diabetes Mellitus in Australia—The Clinical Impact of Changing Population Demographics and Diagnostic Criteria on Prevalence"

_ijerph, 2020, doi:10.3390/ijerph17249387_

Round 1

Reviewer 1 Report

The review written by Laurie et al is interesting and reflects changes that have occurred during the last decade in Australia.

Yet, it is missing the following information:

  1. The authors should add comparative data (paragraph 2 – prevalence) of the rate of GDM in other continents or countries.
  2. Add data regarding the rate of GDMA1 (diet) and GDMA2 (medication) in Australia compared to other continents/countries.
  3. Add information regarding medications used for GDMA2 in Australia and comparison to continents/countries.
  4. COVID-19 – I think it is too early to present conclusions regarding the impact of COVID-19 on GDM. The paragraph about CoVID-19 does not add more information or emphasize the screening program of GDM
  5. Figure 1 should be improved; it is blurry

Author Response

Monday, 7 December 2020

The Editor

Thank you for the informative and constructive comments received from our three reviewers.  Please find below a detailed response to each question raised, with reference made to changes in the body of the text. 

Responses are in Times New Roman and bold italics.

Reviewer 1:

Thank you for the opportunity to address your comments and suggestions.

The authors should add comparative data (paragraph 2 – prevalence) of the rate of GDM in other continents or countries.

Now included (lines 68 - 70)

Add data regarding the rate of GDMA1 (diet) and GDMA2 (medication) in Australia compared to other continents/countries.

We have included available Australian data (line 312) but unfortunately do not have international comparisons.

Add information regarding medications used for GDMA2 in Australia and comparison to continents/countries.

We have included limited available data (line 315) but do not have international comparison for medication use.

COVID-19 – I think it is too early to present conclusions regarding the impact of COVID-19 on GDM. The paragraph about CoVID-19 does not add more information or emphasize the screening program of GDM

The reference to COVID screening practices and potential changes in the numbers of GDM diagnoses in the context of altered screening practices was included to reflect on the topical  and unexpected recommended changes to GDM diagnosis in the context of the COVID-19  pandemic.  We agree that no conclusions can be drawn at this time, but we consider that this issue is sufficiently topical to be included in the current publication.  We have included additional text at lines 344-348 to further clarify this issue.

Figure 1 should be improved figure inserted (line 104).

Reviewer 2 Report

The review by Laurie and McIntyre describes the increases in diagnosed cases of gestational diabetes mellitus in Australia alongside evidence that this increase is not just due to the change in criteria for diagnosing GDM. The authors also highlight key issues with testing and managing GDM specifically in the Australian population.  Overall this review is clear and concise, and the points addressed back up with evidence from published work.

Major comments:

I would like to have seen some comment on whether the changing of the diagnosing system, leading to increased numbers of cases diagnosed, is a positive change.  Does the new criteria mean there are more border line cases being diagnosed and monitored, and is there evidence that there are long term effects of border line levels on the individual and their offspring? Does this take away resources from severe cases? Perhaps some discussion of how the long term effects of this change in criteria could be assessed in the future would give some focus on the future directions of this area.

Minor comments

1) line 116: ‘of’ should be ‘or’

2) line 251: missing reference

Author Response

Monday, 7 December 2020

The Editor

Thank you for the informative and constructive comments received from our three reviewers.  Please find below a detailed response to each question raised, with reference made to changes in the body of the text. 

Responses are in Times New Roman and bold italics.

Reviewer 2:

Thank you for the opportunity to address your comments and suggestions. 

Major comments:

I would like to have seen some comment on whether the changing of the diagnosing system, leading to increased numbers of cases diagnosed, is a positive change. 

This is a difficult issue and remains essentially a matter of opinion rather than one of strict “fact”.  On a population basis, it is difficult to address.  In our article, we have tried to address this in three ways:- 

We note the lack of change in overall neonatal outcomes with retrospective data from Queensland (Ref 23).  This study demonstrated no improvement in overall perinatal outcomes despite the increased diagnoses – from 8.7 – 11.9%.  However, assessing outcomes on a population basis may be difficult as the majority of women (approximately 88% with the higher frequency) are still considered as “non GDM” either before or after the changes are made, so treatment approach actually only differs for the approximate 3.2% of the total population whose status changes from “non GDM” for “GDM” with the changes in diagnostic processes and criteria.  We have added this information in lines 164-169.

We have also discussed the birth outcomes using HAPO data fasting cut offs and lack of change in fetal adverse outcome (Ref 36).

Thirdly, we have mentioned this in conjunction with the COVID screening practice of a fasting level test only (Ref 35). 

Finally (and negatively), we discussed excessive workloads in the section on “models of care” citing thematic analysis of the clinician’s current experience. (Ref 28)

Does the new criteria mean there are more border line cases being diagnosed and monitored, and is there evidence that there are long term effects of border line levels on the individual and their offspring?  

This is a very valid concern, but the available data do not allow us to accurately address this question as GDM diagnoses are recorded only on a Yes / No basis in the larger national surveys.  In long term follow up, the criteria used for initial diagnosis can only be inferred from the time period under consideration.  Included in comments using (Ref 23,35,36)

Does this take away resources from severe cases?

Comments included using (Ref 28)

Perhaps some discussion of how the long term effects of this change in criteria could be assessed in the future would give some focus on the future directions of this area.

Comments included in line 339-343.

Minor comments

line 116: ‘of’ should be ‘or’ Corrected

line 251: missing reference Corrected

Reviewer 3 Report

IJERPH-1014103 presents a review for gestational diabetes. While some parts of the review were interesting, other areas could be improved. I hope the authors consider my feedback.

  • Lines 29-30: A reference should be inserted here to support such a strong statement.
  • Table 1 could be better worked into the text rather than in a table. This information just does not seem to need a table.
  • Line 61: The Cheney et al reference should be directly inserted after “al”. Revise here and where appropriate.
  • Line 64: Should be “(9).”?
  • Figure 1 needs revision because it is not publishable quality. There is no label for the y-axis, the title is included in the figure (it is already in figure title below), and it is blurry.
  • Another nice addition would be to further underscore the need to expand screening such that proper cases are correctly identified. Unidentified cases are of particular concern for healthcare providers and their patients.
  • Could the work in Australia be generalizable to other countries (as a point in the manuscript)?
  • This reviewer was having difficulty trying to identify an epidemiological perspective, as indicated in the title, relative to the clinical and research implications that were more clearly highlighted in the manuscript. Can the authors better include an epidemiological perspective for your paper, or revise the title and keywords appropriately?
  • This type of “review” seems to be a good fit for introducing a theoretical or conceptual model. Is it possible that the authors could work such a model in the manuscript?
  • A large amount of detail for how the articles included in this review needs to be included otherwise the reader will think the articles included in the paper were cherry-picked by the authors. Perhaps you need to be more specific about what type of review this paper is, because it is currently not systematic, narrative, or topical. This is an important point that should be addressed.

Author Response

Monday, 7 December 2020

The Editor

Thank you for the informative and constructive comments received from our three reviewers.  Please find below a detailed response to each question raised, with reference made to changes in the body of the text. 

Responses are in Times New Roman and bold italics.

Reviewer 3:

Thank you for the opportunity to address your comments and suggestions. 

Lines 29-30: A reference should be inserted here to support such a strong statement.

Statement altered.

Table 1 could be better worked into the text rather than in a table. This information just does not seem to need a table.

We consider that it is useful to clearly tabulate the previous and current diagnostic criteria.

Line 61: The Cheney et al reference should be directly inserted after “al”. Revise here and where appropriate .corrected

Line 64: Should be “(9).corrected

Figure 1 needs revision because it is not publishable quality. There is no label for the y-axis, the title is included in the figure (it is already in figure title below), and it is blurry. corrected

Another nice addition would be to further underscore the need to expand screening such that proper cases are correctly identified. Unidentified cases are of particular concern for healthcare providers and their patients.

Please note recommendation for universal testing (line 55) of revised text

Could the work in Australia be generalizable to other countries (as a point in the manuscript)?

We believe that the Australian data are geo specific and likely not generalizable to other countries or regions New comment has been inserted in lines 66 – 70. “There is however, enormous variability between regions in Australia which is discussed further below. International prevalence data drawn from Brown et al(10) in 2017 also shows the high variability of GDM rates ranging from single digit prevalence in Japan, to over 25% of pregnancies affected in California and greater than 45% in the United Arab Emirates.”

This reviewer was having difficulty trying to identify an epidemiological perspective, as indicated in the title, relative to the clinical and research implications that were more clearly highlighted in the manuscript. Can the authors better include an epidemiological perspective for your paper, or revise the title and keywords appropriately?

We have revised the title and key words to better reflect the content of the article.

This type of “review” seems to be a good fit for introducing a theoretical or conceptual model. Is it possible that the authors could work such a model in the manuscript?

We consider that this is a narrative review and follows a pragmatic framework, using available data sources.   We acknowledge that this is not ideal in a strict research context, but we must work within the context of available data.  To clarify, in Australia, these consist of “official” government data (e.g. Australian Institute of Health and Welfare and the “official” perinatal  / “midwifery” data collections which operate on a state by state basis), government supported data from the NDSS (National Diabetes Services Scheme) which provides subsidised care particularly for glucose testing strips and a variety of “local” data sources such as individual clinic databases maintained by various clinical and research groups.  The government data have large numbers included but often sparse detail.  The local datasets contain more detailed information but do not necessarily give nationally representative information.  Only in the Northern Territory (Pandora study as quoted -Ref 24) are these data sources cross checked and validated. 

A large amount of detail for how the articles included in this review needs to be included otherwise the reader will think the articles included in the paper were cherry-picked by the authors. Perhaps you need to be more specific about what type of review this paper is, because it is currently not systematic, narrative, or topical. This is an important point that should be addressed.

As noted, this is a narrative review. We have added details of the method of review included in lines 47 - 53. As noted above, we are limited by available data sources.   Articles were certainly not “cherry picked”, but rather represented available publications.   Text as follows:  “The authors conducted pubmed and medline searches using the terms; GDM, Australia, screening, models of care and postnatal follow up to gather information for this review. Articles were selected to specifically represent Australian cohorts and data for an Australian perspective.  Australian prevalence data was drawn directly from the National Diabetes Service Scheme NDSS (which registers all Australian residents with a diagnosis of gestational diabetes who receive access to government subsidies for the purpose of diabetes care) and the Australian Institute of Health and Welfare website.”

Round 2

Reviewer 1 Report

the authors responded to all comments 

Reviewer 3 Report

The authors have addressed this reviewer's concerns. 

This manuscript is a resubmission of an earlier submission. The following is a list of the peer review reports and author responses from that submission.